# A Deep Conjugate Direction Method for Iteratively Solving Linear Systems

## Abstract

We present a novel deep learning approach to approximate the solution of large, sparse, symmetric, positive-definite linear systems of equations. These systems arise from many problems in applied science, e.g., in numerical methods for partial differential equations. Algorithms for approximating the solution to these systems are often the bottleneck in problems that require their solution, particularly for modern applications that require many millions of unknowns. Indeed, numerical linear algebra techniques have been investigated for many decades to alleviate this computational burden. Recently, data-driven techniques have also shown promise for these problems. Motivated by the conjugate gradients algorithm that iteratively selects search directions for minimizing the matrix norm of the approximation error, we design an approach that utilizes a deep neural network to accelerate convergence via data-driven improvement of the search directions. Our method leverages a carefully chosen convolutional network to approximate the action of the inverse of the linear operator up to an arbitrary constant. We train the network using unsupervised learning with a loss function equal to the $L^2$ difference between an input and the system matrix times the network evaluation, where the unspecified constant in the approximate inverse is accounted for. We demonstrate the efficacy of our approach on spatially discretized Poisson equations with millions of degrees of freedom arising in computational fluid dynamics applications. Unlike state-of-the-art learning approaches, our algorithm is capable of reducing the linear system residual to a given tolerance in a small number of iterations, independent of the problem size. Moreover, our method generalizes effectively to various systems beyond those encountered during training.

## 1 Introduction

In this work, we consider sparse linear systems that arise from discrete Poisson equations in incompressible flow applications (Chorin, 1967; Fedkiw et al., 2001; Bridson, 2008). We use the notation

$$\boldsymbol{Ax} = \boldsymbol{b} \qquad (1)$$

where the dimension $n$ of the matrix $\boldsymbol{A} \in \mathbb{R}^{n \times n}$ and the vector $\boldsymbol{b} \in \mathbb{R}^n$ correlate with spatial fidelity of the computational domain. The appropriate numerical linear algebra technique depends on the nature of the problem. Direct solvers that utilize matrix factorizations (QR, Cholesky, etc. Trefethen & Bau (1997)) have optimal approximation error, but their computational cost is $O(n^3)$ and they typically require dense storage, even for sparse $\boldsymbol{A}$. Although Fast Fourier Transforms (Nussbaumer, 1981) can be used in limited instances (periodic boundary conditions, etc.), iterative techniques are most commonly adopted for these systems given their sparsity. Many applications with strict performance constraints (e.g., real-time fluid simulation) utilize basic iterations (Jacobi, Gauss-Seidel, successive over relaxation (SOR), etc.) given limited computational budget (Saad, 2003). However, large approximation errors must be tolerated since iteration counts are limited by the performance constraints. This is particularly problematic since the wide elliptic spectrum of these matrices (a condition that worsens with increased spatial fidelity/matrix dimension) leads to poor conditioning and iteration counts. Iterative techniques can achieve sub-quadratic convergence if their iteration count does not grow excessively with problem size $n$ since each iteration generally requires $O(n)$ floating point operations for sparse matrices. Discrete elliptic operators are typically symmetric positive (semi) definite and the preconditioned conjugate gradients method (PCG) can be used to

minimize iteration counts (Saad, 2003; Hestenes & Stiefel, 1952; Stiefel, 1952). Preconditioners $P$ for PCG must simultaneously: be symmetric positive definite (SPD) (and therefore admit factorization $P = F^2$), improve the condition number of the preconditioned system $FAFy = Fb$, and be computationally cheap to construct and apply; accordingly, designing specialized preconditioners for particular classes of problems is somewhat of an art. Incomplete Cholesky preconditioners (ICPCG) (Kershaw, 1978) use a sparse approximation to the Cholesky factorization and significantly reduce iteration counts in practice; however, their inherent data dependency prevents efficient parallel implementation. Nonetheless, these are very commonly adopted for Poisson equations arising in incompressible flow (Fedkiw et al., 2001; Bridson, 2008). Multigrid (Brandt, 1977) and domain decomposition (Saad, 2003) preconditioners greatly reduce iterations counts, but they must be updated (with non-trivial cost) each time the problem changes (e.g., in computational domains with time varying boundaries) and/or for different hardware platforms. In general, choice of an optimal preconditioner for discrete elliptic operators is an open area of research.

Recently, data-driven approaches that leverage deep learning techniques have shown promise for solving linear systems. Various researchers have investigated machine learning estimation of multigrid parameters (Greenfeld et al., 2019; Grebhahn et al., 2016; Luz et al., 2020). Others have developed machine learning methods to estimate preconditioners (Götz & Anzt, 2018; Stanaityte, 2020; Ichimura et al., 2020) and initial guesses for iterative methods (Luna et al., 2021; Um et al., 2020; Ackmann et al., 2020). Tompson et al. (2017) and Yang et al. (2016) develop non-iterative machine learning approximations of the inverse of discrete Poisson equations from incompressible flow. We leverage deep learning and develop a novel version of conjugate gradients iterative method for approximating the solution of SPD linear systems which we call the deep conjugate direction method (DCDM). CG iteratively adds $A$-conjugate search directions while minimizing the matrix norm of the error. We use a convolutional neural network (CNN) as an approximation of the inverse of the matrix in order to generate more efficient search directions. We only ask that our network approximate the inverse up to an unknown scaling since this decreases the degree of nonlinearity and since it does not affect the quality of the search direction (which is scale independent). The network is similar to a preconditioner, but it is not a linear function, and our modified conjugate gradients approach is designed to accommodate this nonlinearity. We use unsupervised learning to train our network with a loss function equal to the $L^2$ difference between an input vector and a scaling of $A$ times the output of our network. To account for this unknown scaling during training, we choose the scale of the output of the network by minimizing the matrix norm of the error. Our approach allows for efficient training and generalization to problems unseen (matrices $A$ and right-hand sides $b$). We benchmark our algorithm using the ubiquitous pressure Poisson equation (discretized on regular voxelized domains) and compare against FluidNet (Tompson et al., 2017), which is the state of the art learning-based method for these types of problems. Unlike the non-iterative approaches of Tompson et al. (2017) and Yang et al. (2016), our method can reduce the linear system residuals *arbitrarily*. We showcase our approach with examples that have over 16 million degrees of freedom.

## 2 RELATED WORK

Several papers have focused on enhancing the solution of linear systems (arising from discretized PDEs) using learning. For instance, Götz & Anzt (2018) generate sparsity patterns for block-Jacobi preconditioners using convolutional neural networks, and Stanaityte (2020) use a CNN to predict non-zero patterns for ILU-type preconditioners for the Navier-Stokes equations (though neither work designs fundamentally new preconditioners). Ichimura et al. (2020) develop a neural-network based preconditioner where the network is used to predict approximate Green's functions (which arise in the analytical solution of certain PDEs) that in turn yield an approximate inverse of the linear system. Hsieh et al. (2019) learn an iterator that solves linear systems, performing competitively with classical solvers like multigrid-preconditioned MINRES (Paige & Saunders, 1975). Luz et al. (2020) and Greenfeld et al. (2019) use machine learning to estimate algebraic multigrid (AMG) parameters. They note that AMG approaches rely most fundamentally on effectively chosen (problem-dependent) prolongation sparse matrices and that numerous methods have attempted to automatically create them from the matrix $A$. They train a graph neural network to learn (in an unsupervised fashion) a mapping from matrices $A$ to prolongation operators. Grebhahn et al. (2016) note that geometric multigrid solver parameters can be difficult to choose to guarantee parallel performance on different hardware platforms. They use machine learning to create a code generator to help achieve this.

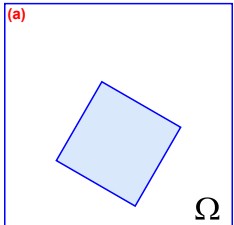 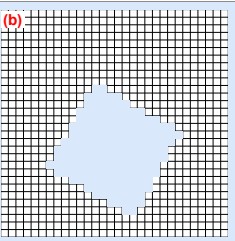 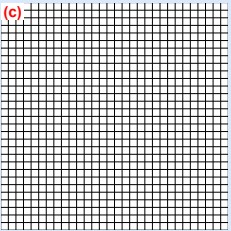 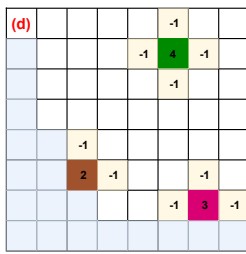

Figure 1: **(a)** We illustrate a sample flow domain $\Omega \subset (0,1)^2$ (in 2D for ease of illustration) with internal boundaries (blue lines). **(b)** We voxelize the domain with a regular grid: white cells represent interior/fluid, and blue cells represent boundary conditions. **(c)** We train using matrix $\boldsymbol{A}^{(0,1)^d}$ from a discretized domain with no interior boundary conditions, where $d$ is the dimension. This creates linear system with $n = (n_c + 1)^d$ unknowns where $n_c$ is the number of grid cells on each direction. **(d)** We illustrate the non-zero entries in an example matrix $\boldsymbol{A}^\Omega$ from the voxelized and labeled (white vs. blue) grid for three example interior cells (green, purple, and brown). Each case illustrates the non-zero entries in the row associated with the example cell. All entries in rows corresponding to boundary/blue cells are zero.

Several works consider accelerating the solution of linear systems by learning an initial guess that is close to the true solution or otherwise helpful to descent algorithms for finding the true solution. In order to solve the discretized Poisson equation, Luna et al. (2021) accelerate the convergence of GMRES (Saad & Schultz, 1986) with an initial guess that is learned in real-time (i.e., as a simulation code runs) with no prior data. Um et al. (2020) train a network (incorporating differentiable physics, based on the underlying PDEs) in order to produce high-quality initial guesses for a CG solver. In a somewhat similar vein, Ackmann et al. (2020) use a simple feedforward neural network to predict pointwise solution components, which accelerates the conjugate residual method used to solve a relatively simple shallow-water model (a more sophisticated network and loss function are needed to handle more general PDEs and larger-scale problems).

At least two papers (Ruelmann et al., 2018; Sappl et al., 2019) have sought to learn a mapping between a matrix and an associated sparse approximate inverse. In their investigation, Ruelmann et al. (2018) propose training a neural network using matrix-inverse pairs as training data. Although straightforward to implement, the cost of generating training data, let alone training the network, is prohibitive for large-scale 3D problems. Sappl et al. (2019) seek to learn a mapping between linear system matrices and sparse (banded) approximate inverses. Their loss function is the condition number of the product of the system matrix and the approximate inverse; the minimum value of the condition number is one, which is achieved exactly when an exact inverse is obtained. Although this framework is quite simple, evaluating the condition number of a matrix is asymptotically costly, and in general, the inverse of a sparse matrix can be quite dense. Accordingly, the method is not efficient or accurate enough for the large-scale 3D problems that arise in real-world engineering problems.

Most relevant to the present work is FluidNet (Tompson et al., 2017). FluidNet develops a highly-tailored CNN architecture that is used to predict the solution of a linear projection operation (specifically, for the discrete Poisson equation) given a matrix and right-hand side. The authors demonstrate fluid simulations where the linear solve is replaced by evaluating their network. Because their network is relatively lightweight and is only evaluated once per time step, their simulations run efficiently. However, their design allows the network only one opportunity to reduce the residual for the linear solve; in practice, we observe that FluidNet is able to reduce the residual by no more than about one order of magnitude. However, in computer graphics applications, at least four orders of magnitude in residual reduction are usually required for visual fidelity, while in scientific and engineering applications, practitioners prefer solutions that reduce the residual by eight or more orders of magnitude (i.e., to within machine precision). Accordingly, FluidNet's lack of convergence stands in stark contrast to classical, convergent methods like CG. Our method resolves this gap.

## 3 Motivation: Incompressible Flow

We demonstrate the efficacy of our approach with the linear systems that arise in incompressible flow applications. Specifically, we use our algorithm to solve the discrete Poisson equations in

regular-grid-based discretization of the pressure projection equations that arise in Chorin's splitting technique (Chorin, 1967) for the inviscid, incompressible Euler equations. These equations are

$$\rho \left( \frac{\partial \boldsymbol{u}}{\partial t} + \frac{\partial \boldsymbol{u}}{\partial \boldsymbol{x}} \boldsymbol{u} \right) + \nabla p = \boldsymbol{f}^{ext}, \qquad \nabla \cdot \boldsymbol{u} = 0 \tag{2}$$

where $\boldsymbol{u}$ is fluid velocity, $p$ is pressure, $\rho$ is density, and $\boldsymbol{f}^{ext}$ accounts for external forces like gravity. The equations are assumed at all positions $\boldsymbol{x}$ in the spatial fluid flow domain $\Omega$ and for time $t > 0$. The left term in Equation 2 enforces conservation of momentum in the absence of viscosity, and the second part enforces incompressibility and conservation of mass. These equations are subject to initial conditions $\rho(\boldsymbol{x}, 0) = \rho^0$ and $\boldsymbol{u}(\boldsymbol{x}, 0) = \boldsymbol{u}^0(\boldsymbol{x})$ as well as boundary conditions $\boldsymbol{u}(\boldsymbol{x}, t) \cdot \boldsymbol{n}(\boldsymbol{x}) = u^{\partial \Omega}(\boldsymbol{x}, t)$ on the boundary of the domain $\boldsymbol{x} \in \partial \Omega$ (where $\boldsymbol{n}$ is the unit outward pointing normal at position $\boldsymbol{x}$ on the boundary). Equation 2 is discretized in both time and space. Temporally, we split the advection $\frac{\partial \boldsymbol{u}}{\partial t} + \frac{\partial \boldsymbol{u}}{\partial \boldsymbol{x}} \boldsymbol{u} = 0$ and body forces terms $\rho \frac{\partial \boldsymbol{u}}{\partial t} = \boldsymbol{f}^{ext}$, and finally enforce incompressibility via the pressure projection $\frac{\partial \boldsymbol{u}}{\partial t} + \frac{1}{\rho} \nabla p = \boldsymbol{0}$ such that $\nabla \cdot \boldsymbol{u} = 0$; this is the standard advection-projection scheme proposed by Chorin (1967). Using finite differences in time, we can summarize this as

$$\rho^0 \left( \frac{\boldsymbol{u}^* - \boldsymbol{u}^n}{\Delta t} + \frac{\partial \boldsymbol{u}^n}{\partial \boldsymbol{x}} \boldsymbol{u}^n \right) = \boldsymbol{f}^{ext} \tag{3}$$

$$-\nabla \cdot \frac{1}{\rho^0} \nabla p^{n+1} = -\nabla \cdot \boldsymbol{u}^* \tag{4}$$

$$-\frac{1}{\rho^0} \nabla p^{n+1} \cdot \boldsymbol{n} = \frac{1}{\Delta t} \left( u^{\partial \Omega} - \boldsymbol{u}^* \cdot \boldsymbol{n} \right). \tag{5}$$

For the spatial discretization, we use a regular marker-and-cell (MAC) grid (Harlow & Welch, 1965) with cubic voxels whereby velocity components are stored on the face of voxel cells, and scalar quantities (e.g., pressure $p$ or density $\rho$) are stored at voxel centers. We use backward semi-Lagrangian advection (Fedkiw et al., 2001; Gagniere et al., 2020) for Equation 3. All spatial partial derivatives are approximated using finite differences. Equations 4 and 5 describe the pressure Poisson equation with Neumann conditions on the boundary of the flow domain. We discretize the left hand side of Equation 4 using a standard 7-point finite difference stencil. The right-hand side is discretized using the MAC grid discrete divergence finite difference stencils as well as contributions from the boundary condition terms in Equation 5. We refer the reader to Bridson (2008) for more in-depth implementation details. Equation 5 is discretized by modifying the Poisson stencil to enforce Neumann boundary conditions. We do this using a simple labeling of the voxels in the domain. For simplicity, we assume $\Omega \subset (0, 1)^3$ is a subset of the unit cube, potentially with internal boundaries. We label cells in the domain as either liquid or boundary. This simple classification is enough to define the Poisson discretizations (with appropriate Neumann boundary conditions at domain boundaries) that we focus on in the present work; we illustrate the details in Figure 1. We use the following notation to denote the discrete Poisson equations associated with Equations 4–5:

$$\boldsymbol{A}^\Omega \boldsymbol{x} = \boldsymbol{b}^{\nabla \cdot \boldsymbol{u}^*} + \boldsymbol{b}^{u^{\partial \Omega}}, \tag{6}$$

where $\boldsymbol{A}^\Omega$ is the discrete Poisson matrix associated with the voxelized domain, $\boldsymbol{x}$ is the vector of unknown pressure, and $\boldsymbol{b}^{\nabla \cdot \boldsymbol{u}^*}$ and $\boldsymbol{b}^{u^{\partial \Omega}}$ are the right-hand side terms from Equations 4 and 5, respectively. We define a special case of the matrix involved in this discretization to be the Poisson matrix $\boldsymbol{A}^{(0,1)^3}$ associated with $\Omega = (0, 1)^3$, i.e., a full fluid domain with no internal boundaries. We use this matrix for training, yet demonstrate that our network generalizes to all other matrices arising from more complicated flow domains.

## 4 DEEP CONJUGATE DIRECTION METHOD

We present our method for the deep learning acceleration of iterative approximations to the solution of linear systems of the form seen in Equation 6. We first discuss relevant details of the conjugate gradients (CG) method, particularly line search and $\boldsymbol{A}$-orthogonal search directions. We then present a deep learning technique for improving the quality of these search directions that ultimately reduces

iteration counts required to achieve satisfactory residual reduction. Lastly, we outline the training procedures for our deep convolutional neural network.

Our approach iteratively improves approximations to the solution $\boldsymbol{x}$ of Equation 6. We build on the method of CG, which requires the matrix $\boldsymbol{A}^\Omega$ in Equation 6 to be SPD. SPD matrices $\boldsymbol{A}^\Omega$ give rise to the matrix norm $\|\boldsymbol{y}\|_{\boldsymbol{A}^\Omega} = \sqrt{\boldsymbol{y}^T \boldsymbol{A}^\Omega \boldsymbol{y}}$. CG can be derived in terms of iterative line search improvement based on optimality in this norm. That is, an iterate $\boldsymbol{x}_{k-1} \approx \boldsymbol{x}$ is updated in search direction $\boldsymbol{d}_k$ by a step size $\alpha_k$ that is chosen to minimize the matrix norm of the error:

$$\alpha_k = \arg\min_\alpha \frac{1}{2} \|\boldsymbol{x} - (\boldsymbol{x}_{k-1} + \alpha \boldsymbol{d}_k)\|_{\boldsymbol{A}^\Omega}^2 = \frac{\boldsymbol{r}_{k-1}^T \boldsymbol{d}_k}{\boldsymbol{d}_k^T \boldsymbol{A}^\Omega \boldsymbol{d}_k}, \tag{7}$$

where $\boldsymbol{r}_{k-1} = \boldsymbol{b} - \boldsymbol{A}^\Omega \boldsymbol{x}_{k-1}$ is the $(k-1)^{\text{th}}$ residual and $\boldsymbol{b}$ is the right-hand side in Equation 6. Different search directions result in different algorithms. A natural choice is the negative gradient of the matrix norm of the error (evaluated at the current iterate), since this will point in the direction of steepest decrease $\boldsymbol{d}_k = -\frac{1}{2}\nabla \|\boldsymbol{x}_{k-1}\|_{\boldsymbol{A}^\Omega}^2 = \boldsymbol{r}_{k-1}$. This is the gradient descent method (GD). Unfortunately, this approach requires many iterations in practice. A more effective strategy is to choose directions that are $\boldsymbol{A}$-orthogonal (i.e., $\boldsymbol{d}_i^T \boldsymbol{A}^\Omega \boldsymbol{d}_j = 0$ for $i \neq j$). With this choice, the search directions form a basis for $\mathbb{R}^n$ so that the initial error can be written as $\boldsymbol{e}_0 = \boldsymbol{x} - \boldsymbol{x}_0 = \sum_{i=1}^n e_i \boldsymbol{d}_i$, where $e_i$ are the components of the initial error written in the basis. Furthermore, when the search directions are $\boldsymbol{A}$-orthogonal, the optimal step sizes $\alpha_k$ at each iteration satisfy

$$\alpha_k = \frac{\boldsymbol{r}_{k-1}^T \boldsymbol{d}_k}{\boldsymbol{d}_k^T \boldsymbol{A}^\Omega \boldsymbol{d}_k} = \frac{\boldsymbol{d}_k^T \boldsymbol{A}^\Omega \boldsymbol{e}_{k-1}}{\boldsymbol{d}_k^T \boldsymbol{A}^\Omega \boldsymbol{d}_k} = \frac{\boldsymbol{d}_k^T \boldsymbol{A}^\Omega \left(\sum_{i=1}^n e_i \boldsymbol{d}_i - \sum_{j=1}^{k-1} \alpha_j \boldsymbol{d}_j\right)}{\boldsymbol{d}_k^T \boldsymbol{A}^\Omega \boldsymbol{d}_k} = e_k.$$

That is, the optimal step sizes are chosen to precisely eliminate the components of the error on the basis defined by the search directions. Thus, convergence is determined by the (at most $n$) non-zero components $e_i$ in the initial error. Although rounding errors prevent this from happening exactly in practice, this property greatly reduces the number of required iterations (Golub & Loan, 2012).

CG can be viewed as a modification of GD where the search direction is chosen as the component of the residual (equivalently, the negative gradient of the matrix norm of the error) that is $\boldsymbol{A}$-orthogonal to all previous search directions:

$$\boldsymbol{d}_k = \boldsymbol{r}_{k-1} - \sum_{i=1}^{k-1} h_{ik} \boldsymbol{d}_i, \qquad h_{ik} = \frac{\boldsymbol{d}_i^T \boldsymbol{A}^\Omega \boldsymbol{r}_{k-1}}{\boldsymbol{d}_i^T \boldsymbol{A}^\Omega \boldsymbol{d}_i}.$$

In practice, $h_{ik} = 0$ for $i < k - 1$, and this iteration can therefore be performed without the need to store all previous search directions $\boldsymbol{d}_i$ and without the need for computing all previous $h_{ik}$.

While the residual is a natural choice for generating $\boldsymbol{A}$-orthogonal search directions (since it points in the direction of the steepest local decrease), it is not the optimal search direction. If $\boldsymbol{d}_k$ is parallel to $(\boldsymbol{A}^\Omega)^{-1}\boldsymbol{r}_{k-1}$, then $\boldsymbol{x}_k$ will be equal to $\boldsymbol{x}$ since $\alpha_k$ (computed from Equation 7) will step directly to the solution. We can see this by considering the residual and its relation to the search direction:

$$\boldsymbol{r}_k = \boldsymbol{b} - \boldsymbol{A}^\Omega \boldsymbol{x}_k = \boldsymbol{b} - \boldsymbol{A}^\Omega \boldsymbol{x}_{k-1} - \alpha_k \boldsymbol{A}^\Omega \boldsymbol{d}_k = \boldsymbol{r}_{k-1} - \alpha_k \boldsymbol{A}^\Omega \boldsymbol{d}_k.$$

In light of this, we use deep learning to create an approximation $\boldsymbol{f}(\boldsymbol{c}, \boldsymbol{r})$ to $(\boldsymbol{A}^\Omega)^{-1}\boldsymbol{r}$, where $\boldsymbol{c}$ denotes the network weights and biases. This is analogous to using a preconditioner in PCG; however, our network is not SPD (nor even a linear function). We simply use this data-driven approach as our means of generating better search directions $\boldsymbol{d}_k$. Furthermore, we only need to approximate a vector parallel to $(\boldsymbol{A}^\Omega)^{-1}\boldsymbol{r}$ since the step size $\alpha_k$ will account for any scaling in practice. In other words, $\boldsymbol{f}(\boldsymbol{c}, \boldsymbol{r}) \approx s_{\boldsymbol{r}} (\boldsymbol{A}^\Omega)^{-1}\boldsymbol{r}$, where the scalar $s_{\boldsymbol{r}}$ is not defined globally; it only depends on $\boldsymbol{r}$, and the model does not learn it. Lastly, as with CG, we enforce $\boldsymbol{A}$-orthogonality, yielding search directions

$$\boldsymbol{d}_k = \boldsymbol{f}(\boldsymbol{c}, \boldsymbol{r}_{k-1}) - \sum_{i=1}^{k-1} h_{ik} \boldsymbol{d}_i, \qquad h_{ik} = \frac{\boldsymbol{f}(\boldsymbol{c}, \boldsymbol{r}_{k-1})^T \boldsymbol{A}^\Omega \boldsymbol{d}_i}{\boldsymbol{d}_i^T \boldsymbol{A}^\Omega \boldsymbol{d}_i}.$$

We summarize our approach in Algorithm 1. Note that we introduce the variable $i_{\text{start}}$. To guarantee $\boldsymbol{A}$-orthogonality between all search directions, we must have $i_{\text{start}} = 1$. However, this requires storing all prior search directions, which can be costly. We found that using $i_{\text{start}} = k - 2$ worked nearly as well as $i_{\text{start}} = 1$ in practice (in terms of our ability to iteratively reduce the residual of the system). We demonstrate this in Figure 4c.

## 5 MODEL ARCHITECTURE, DATASETS, AND TRAINING

Efficient performance of our method requires effective training of our deep convolutional network for weights and biases $c$ such that $f(c, r) \approx s_r(A^\Omega)^{-1}r$ (for arbitrary scalar $s_r$). We design a model architecture, loss function, and unsupervised training approach to achieve this. Our approach has modest training requirements and allows for effective residual reduction while generalizing well to problems not seen in the training data.

**Algorithm 1 DCDM**

---
1: $r_0 = b - A^\Omega x_0$
2: $k = 1$
3: **while** $\|r_{k-1}\| \geq \epsilon$ **do**
4:     $d_k = f(c, \frac{r_{k-1}}{\|r_{k-1}\|})$
5:     **for** $i_{\text{start}} \leq i < k$ **do**
6:         $h_{ik} = \frac{d_k^T A^\Omega d_i}{d_i^T A^\Omega d_i}$
7:         $d_k \mathrel{-}= h_{ik}d_i$
8:     **end for**
9:     $\alpha_k = \frac{r_{k-1}^T d_k}{d_k^T A^\Omega d_k}$
10:     $x_k = x_{k-1} + \alpha_k d_k$
11:     $r_k = b - A^\Omega x_k$
12:     $k = k + 1$
13: **end while**

---

### 5.1 LOSS FUNCTION AND UNSUPERVISED LEARNING

Although we generalize to arbitrary matrices $A^\Omega$ from Equation 6 that correspond to domains $\Omega \subset (0,1)^3$ that have internal boundaries (see Figure 1), we train using just the matrix $A^{(0,1)^3}$ from the full cube domain $(0,1)^3$. In contrast, other similar approaches (Tompson et al., 2017; Yang et al., 2016) train using matrices $A^\Omega$ and right-hand sides $b^{\nabla \cdot u^*} + b^{u^{\partial \Omega}}$ that arise from flow in many domains with internal boundaries. We train our network by minimizing the $L^2$ difference $\|r - \alpha A^{(0,1)^3} f(c, r)\|_2$, where $\alpha = \frac{r^T f(c,r)}{f(c,r)^T A^{(0,1)^3} f(c,r)}$ from Equation 7. This choice of $\alpha$ accounts for the unknown scaling in the approximation of $f(c, r)$ to $\left(A^{(0,1)^3}\right)^{-1}r$. We use an unsupervised approach and train the model by minimizing

$$\text{Loss}(f, c, \mathcal{D}) = \frac{1}{|\mathcal{D}|}\sum_{r \in \mathcal{D}}\|r - \frac{r^T f(c,r)}{f(c,r)^T A^{(0,1)^3} f(c,r)}A^{(0,1)^3} f(c, r)\|_2$$

for given dataset $\mathcal{D}$ consisting of training vectors $b^i$. In Algorithm 1, the normalized residuals $\frac{r_k}{\|r_k\|}$ are passed as inputs to the model. Unlike in e.g. FluidNet (Tompson et al., 2017), only the first residual $\frac{r_0}{\|r_0\|}$ is directly related to the problem-dependent original right-hand side $b$. Hence we consider a broader range of training vectors than those expected in a given problem of interest, e.g., incompressible flows. We observe that generally the residuals $r_k$ in Algorithm 1 are skewed to the lower end of the spectrum of the matrix $A^\Omega$. Since $A^\Omega$ is a discretized elliptic operator, lower end modes are of lower frequency of spatial oscillation. We create our training vectors $b^i \in \mathcal{D}$ using $m \ll n$ approximate eigenvectors of the training matrix $A^{(0,1)^3}$. We use the Rayleigh-Ritz method to create approximate eigenvectors $q_i$, $0 \leq i < m$. This approach allows us to effectively approximate the full spectrum of $A^{(0,1)^3}$ without computing the full eigendecomposition, which can be expensive ($O(n^3)$) at high resolution. We found that using $m = 10000$ worked well in practice. The Rayleigh-Ritz vectors are orthonormal and satisfy $Q_m^T A^{(0,1)^3} Q_m = \Lambda_m$, where $\Lambda_m$ is a diagonal matrix with nondecreasing diagonal entries $\lambda_i$ referred to as Ritz values (approximate eigenvalues) and $Q_m = [q_0, q_1, \ldots, q_{m-1}] \in \mathbb{R}^{n \times m}$. We pick $b^i = \frac{\sum_{j=0}^{m-1} c_j^i q_j}{\|\sum_{j=0}^{m-1} c_j^i q_j\|}$, where the coefficients $c_j^i$ are picked from a standard normal distribution

$$c_j^i = \begin{cases} 9 \cdot \mathcal{N}(0,1) & \text{if } \tilde{j} \leq j \leq \frac{m}{2} + \theta \\ \mathcal{N}(0,1) & \text{otherwise} \end{cases}$$

where $\theta$ is a small number (we used $\theta = 500$), and $\tilde{j}$ is the first index that $\lambda_{\tilde{j}} > 0$. This choice creates 90% of $b^i$ from the lower end of the spectrum, with the remaining 10% from the higher end. The Riemann-Lebesgue Lemma states the Fourier spectrum of a continuous function will decay at infinity, so this specific choice of $b_i$'s is reasonable for the training set. In practice, we also observed that the right-hand sides of the pressure system that arose in flow problems (in the empty domain) tended to be at the lower end of the spectrum. Notably, even though this dataset only uses Rayleigh-Ritz vectors from the training matrix $A^{(0,1)^3}$, our network can be effectively generalized to flows in irregular domains, e.g., smoke flow past a rotating box and flow past a bunny (see Figure 3).

We generate the Rayleigh-Ritz vectors by first tridiagonalizing the training matrix $A^{(0,1)^3}$ with $m$ Lanczos iterations (Lanczos, 1950) to form $T^m = Q_m^{L}{}^T A^{(0,1)^3} Q_m^L \in \mathbb{R}^{m \times m}$. We then diagonalize

$\boldsymbol{T}^m = \hat{\boldsymbol{Q}}^T \boldsymbol{\Lambda}_m \hat{\boldsymbol{Q}}$. While costly, we note that this algorithm is performed on the comparably small $m \times m$ matrix $\boldsymbol{T}^m$ (rather than on the $\boldsymbol{A}^{(0,1)^3} \in \mathbb{R}^{n \times n}$). This yields the Rayleigh-Ritz vectors as the columns of $\boldsymbol{Q}_m = \boldsymbol{Q}_m^L \hat{\boldsymbol{Q}}$. The Lanczos vectors are the columns of the matrix $\boldsymbol{Q}_m^L$ and satisfy a three-term recurrence whereby the next Lanczos vector can be computed from the previous two as

$$\beta_j \boldsymbol{q}_{j+1}^L = \boldsymbol{A}^{(0,1)^3} \boldsymbol{q}_j^L - \beta_{j-1} \boldsymbol{q}_{j-1}^L - \alpha_j \boldsymbol{q}_j^L,$$

where $\alpha_j$ and $\beta_j$ are diagonal and subdiagonal entries of $\boldsymbol{T}^k$. $\beta_j$ is computed so that $\boldsymbol{q}_{j+1}^L$ is a unit vector, and $\alpha_{j+1} = \boldsymbol{q}_{j+1}^T \boldsymbol{A}^{(0,1)^3} \boldsymbol{q}_{j+1}$. We initialize the iteration with a random $\boldsymbol{q}_0^L \in \text{span}(\boldsymbol{A}^{(0,1)^3})$. The Lanczos algorithm can be viewed as a modified Gram-Schmidt technique to create an orthonormal basis for the Krylov space associated with $\boldsymbol{q}_0^L$ and $\boldsymbol{A}^{(0,1)^3}$, and it therefore suffers from rounding error sensitivities manifested as loss of orthonormality with vectors that do not appear in the recurrence. We found that the simple strategy described in Paige (1971) of orthogonalizing each iterate with all previous Lanczos vectors to be sufficient for our training purposes. Dataset creation takes 5–7 hours for a $64^3$ computational grid, and 2–2.5 days for a $128^3$ grid.

## 5.2 Model Architecture

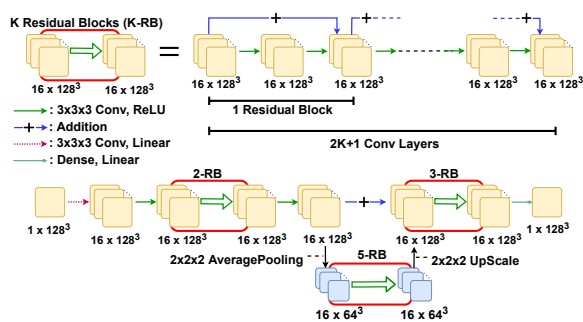

Figure 2: Model architecture for training with $\boldsymbol{A}^{(0,1)^3}$ on a $128^3$ grid.

The internal structure of our CNN architecture for a $128^3$ grid is shown in Figure 2. It consists of a series of convolutional layers with residual connections. The upper left of Figure 2 ($K$ Residual Blocks) shows our use of multiple blocks of residually connected layers. Notably, within each block, the first layer directly affects the last layer with an addition operator. All non-input or output convolutions use a $3 \times 3 \times 3$ filter, and all layers consist of 16 feature maps. In the middle of the first level, a layer is downsampled (via the average pooling operator with $(2 \times 2 \times 2)$ pool size) and another set of convolutional layers is applied with residual connection blocks. The last layer in the second level is upscaled and added to the layer that is downsampled. The last layer in the network is dense with a linear activation function. The activation functions in all convolutional layers are ReLU, except for the first convolution, which uses a linear activation function.

Initially we tried a simple deep feedforward convolutional network with residual connections (motivated by He et al. (2016)). Although such a simple model works well for DCDM, it requires high number of layers, which results in higher training and inference times. We found that creating parallel layers of CNNs with downsampling reduced the number of layers required. In summary, our goal was to first identify the simplest network architecture that provided adequate accuracy for our target problems, and subsequently, we sought to make architectural changes to minimize training and inference time; further optimizations are possible.

Differing resolutions use differing numbers of convolutions, but the fundamental structure remains the same. More precisely, the number of residual connections is changed for different resolutions. For example, a $64^3$ grid uses one residual block on the left, two on the right on the upper level, and three on the lower level. Furthermore, the weights trained on a lower resolution grid can be used effectively with higher resolutions. Figure 4d shows convergence results for a $256^3$ grid, using a model trained for a $64^3$ grid and a $128^3$ grid. The model that we use for $256^3$ grids in our final examples was trained on a $128^3$ grid; however, as the shown in the figure, even training with a $64^3$ grid allows for efficient residual reduction. Table 1 shows results for three different resolutions, where DCDM uses $64^3$ and $128^3$ trained models. This approach makes the number of parameters in the model independent of the spatial fidelity of the problem.

## 5.3 TRAINING

Using the procedure explained in Section 5.1, we create the training dataset $\mathcal{D} \in \text{span}(\boldsymbol{A}^{(0,1)^3}) \cap \mathcal{S}^{n-1}$ of size 20000 generated from 10000 Rayleigh-Ritz vectors. We train our model with TensorFlow (Abadi et al., 2015) on a single NVIDIA RTX A6000 GPU with 48GB memory. Training is done with standard deep learning techniques—more precisely, back-propagation and the ADAM optimizer (Kingma & Ba, 2015) (with starting learning rate 0.0001). Training takes approximately 10 minutes and 1 hour per epoch for grid resolutions $64^3$ and $128^3$, respectively. We trained our model for 50 epochs; however, the model from the thirty-

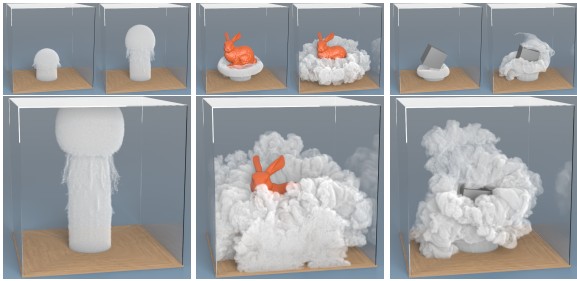

Figure 3: DCDM for simulating a variety of incompressible flow examples. Left: smoke plume at $t = 6.67, 13.33, 20$ seconds. Middle: smoke passing bunny at $t = 5, 10, 15$ seconds. Right: smoke passing a spinning box (time-dependent Neumann boundary conditions) at $t = 2.67, 6, 9.33$ seconds.

first epoch was optimal for $64^3$, and the model from the third epoch was optimal for $128^3$.

## 6 RESULTS AND ANALYSIS

We demonstrate DCDM on three increasingly difficult examples and provide numerical evidence for the efficient convergence of our method. All examples were run on a workstation with dual AMD EPYC 75F3 processors and 512GB RAM.

Figure 3 showcases DCDM for incompressible smoke simulations. In each simulation, inlet boundary conditions are set in a circular portion of the bottom of the cubic domain, whereby smoke flows around potential obstacles and fills the domain. We show a smoke plume (no obstacles), flow past a complex static geometry (the Stanford bunny), and flow past a dynamic geometry (a rotating cube). Visually plausible and highly-detailed results are achieved for each simulation (see supplementary material for larger videos). The plume example uses a computational grid with resolution $128^3$, while the other two uses grids with resolution $256^3$ (representing over 16 million unknowns). For each linear solve, DCDM was run until the residual was reduced by four orders of magnitude.

For the bunny example, Figures 4a–b demonstrate how residuals decrease over the course of a linear solve, comparing DCDM with other methods. Figure 4a shows the mean results (with standard deviations) over the course of 400 simulation frames, while in Figure 4b, we illustrate behavior on a particular frame (frame 150). For

Table 1: Timing and iteration comparison for different methods on the bunny example. DCDM-{64,128} calls a model whose parameters trained over a $\{64^3, 128^3\}$ grid. All computations are done using only CPUs; model inference does not use GPUs.

| Method | $64^3$ Grid $t_r$ | $64^3$ Grid $n_r$ | $128^3$ Grid $t_r$ | $128^3$ Grid $n_r$ | $256^3$ Grid $t_r$ | $256^3$ Grid $n_r$ |
|---|---|---|---|---|---|---|
| DCDM-64 | 2.71s | **16** | **22s** | 27 | **261s** | 58 |
| DCDM-128 | 5.37s | 19 | 26s | **24** | 267s | **44** |
| CG | **1.77**s | 168 | 26s | 465 | 1548s | 1046 |
| Deflated PCG | 771.6s | 117 | 3700s | 277 | 21030s | 489 |

FluidNet, we use the implementation provided by fluidnetsc22 (2022). This implementation includes pre-trained models that we use without modification. In both subfigures, it is evident that the FluidNet residual never changes, since the method is not iterative; FluidNet reduces the initial residual by no more than one order of magnitude. On the other hand, with DCDM, we can continually reduce the residual (e.g., by four orders of magnitude) as we apply more iterations of our method, just as with classical CG. In Figure 4b, we also visualize the convergence of three other classical methods, CG, Deflated PCG (Saad et al., 2000), and incomplete Cholesky preconditioned CG (ICPCG)); clearly, DCDM reduces the residual in the fewest number of iterations (e.g., approximately one order of magnitude fewer iterations than ICPCG). Since FluidNet is not an

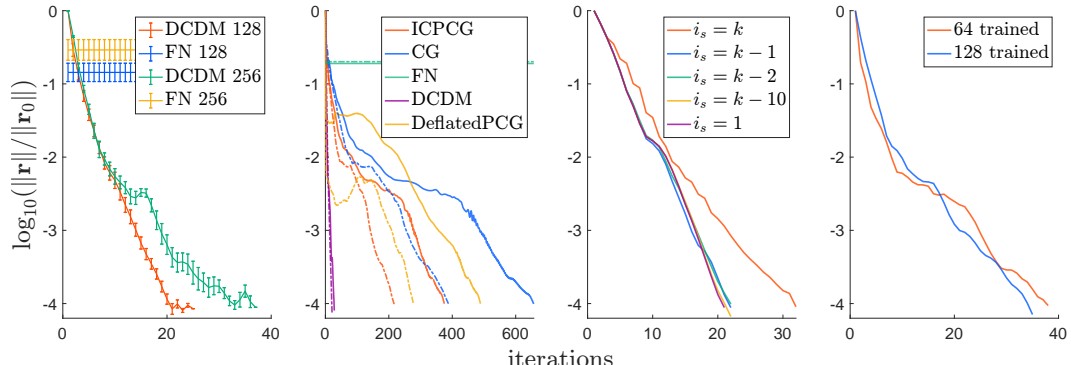

Figure 4: Convergence data for the bunny example. **(a)** Mean and std. dev. (over all 400 frames in the simulation) of residual reduction during linear solves (with $128^3$ and $256^3$ grids) using FluidNet (FN) and DCDM. **(b)** Residual plots with CG, ICPCG, Deflated PCG, FN, and DCDM at frame 150. Dashed and solid lines represent results for $128^3$ and $256^3$, respectively. **(c)** Decrease in residuals with varying degrees of $\boldsymbol{A}$-orthogonalization ($i_s = i_{\text{start}}$). **(d)** Reduction in residuals when the network is trained with a $64^3$ or $128^3$ grid for the $256^3$ grid simulation shown in Figure 3 Middle.

iterative method and lacks a notion of residual reduction, we treat $\boldsymbol{r}_0$ for FluidNet as though an initial guess of zero is used (as is done in our solver).

To clarify these results, Table 1 reports convergence statistics for DCDM compared to standard iterative techniques CG and Deflated PCG. For all $64^3$, $128^3$, and $256^3$ grids with the bunny example, we measure the time $t_r$ and the number of iterations $n_r$ required to reduce the initial residual on a particular time step of the simulation by four orders of magnitude. DCDM achieves the desired results in by far the fewest number of iterations at all resolutions. At $256^3$, DCDM performs approximately 6 times faster than CG, suggesting a potentially even wider performance advantage at higher resolutions. Inference is the dominant cost in an iteration of DCDM; the other linear algebra computations in an iteration of DCDM are comparable to those in CG. The nice result of our method is that despite the increased time per iteration, the number of required iterations is reduced so drastically that DCDM materially outperforms classical methods like CG. Although ICPCG successfully reduces number of iterations 4 b, we found the runtime to scale prohibitively with grid resolution, so we exclude it from comparison in table 1. Notably, even though Deflated PCG and DCDM are based on approximate Ritz vectors, DCDM performs far better, indicating the value of using a neural network.

## 7 Conclusions

We presented DCDM, incorporating CNNs into a CG-style algorithm that yields efficient, convergent behavior for solving linear systems. Our method is evaluated on linear systems with over 16 million degrees of freedom and converges to a desired tolerance in merely tens of iterations. Furthermore, despite training the underlying network on domains without obstacles, our network is able to successfully predict search directions that enable efficient linear solves on domains with complex and dynamic geometries. Moreover, the training data for our network does not require running fluid simulations or solving linear systems ahead of time; our Rayleigh-Ritz vector approach enables us to quickly generate very large training datasets, unlike approaches seen in other works.

Our network was designed for and trained exclusively using data related to the discrete Poisson matrix, which likely limits the generalizability of our present model. However, we believe our *method* is readily applicable to other classes of PDEs (or general problems with graph structure) that give rise to large, sparse, symmetric linear systems. We note that our method is unlikely to work well for matrices that have high computational cost to evaluate $\boldsymbol{A} * x$ (such as dense matrices), since training relies on efficient $\boldsymbol{A} * \boldsymbol{x}$ evaluations. An interesting question to consider is how well our method and current models would apply to discrete Poisson matrices arising from non-uniform grids, e.g., quadtrees or octrees (Losasso et al., 2004).

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
