# OpenReview forum: "A Deep Conjugate Direction Method for Iteratively Solving Linear Systems"
_ICLR.cc/2023/Conference — Submitted to ICLR 2023_

### Official Review · Reviewer_gFH2 · 2022-10-19

**Confidence:** 3
**Correctness:** 3
**Technical Novelty And Significance:** 3
**Empirical Novelty And Significance:** 4
**Recommendation:** 3

**Clarity, Quality, Novelty And Reproducibility:**

Clarity: The paper is clearly written but could benefit from a shift in emphasis. For example instead of explaining the details of the specific PDE from which the matrix is generated it could be more useful to discuss the type of structure that results from such PDEs, i.e. what does the sparsity pattern look like, what does the pattern of the inverse look like, plot the matrix elements as a distribution to see smoothness properties etc.

Quality: The paper is somewhat limited in it's scope and quality could be improved significantly by a) generalizing to more example use cases when trying to keep it an experimental paper and b) analyzing the theoretical considerations of the approach such as for example how much can the network even learn to help reduce the residual given a certain number of m eigenvectors for training

Novelty: I'm not sufficiently familiar with the literature in order to judge the novelty of the presented approach.

Reproducibility: I don't see any references to the code but the method itself is explained well enough to facilitate reproducibility in principle.


**Details Of Ethics Concerns:**

No concerns

**Strength And Weaknesses:**

Strengths:
- The paper does a good job explaining the specific problem setup.
- The paper indicates that there is a possible generalization effect by scaling the grid size for a given problem instance and this could be explored further.

Weaknesses:
- The paper addresses a very specific problem type and is not as general as the title suggests.
- For someone not familiar with the literature on approximating matrix inverses the paper could improve on motivation for the network architecture, for example why are non-linearities even necessary / helful if the learned function is linear.
- The paper does not contain a theoretical analysis of the computational complexity of obtaining eigenvalue and matrix inverse approximations. To me it is not clear why a method that uses stochastic gradients (Adam) should converge to a good matrix inverse approximation faster than more specific deterministic approaches, especially if you already have a sample of eigenvectors available.
- The experimental setup is not always well motivated and seems somewhat arbitraty, for example it seems to me that the choice of m=10k example eigenvectors should be somehow related to the eigenspectrum.
- The paper does not consider the computational cost of the whole end-to-end setup. Is the method useful if only a single solve with the matrix A is performed given how expensive it is to generate the training set and train the network? If we require multiple solves with the e.g. similar matrices A, then how many solves are typical for a given PDE solver and how useful can this method be? What residual norm for the linear system is required in order for the PDE solver to converge?
- The paper does not show how well the network model actually learns / generalizes on a validation set and does not compare different architecture options in that way.
- I'm not sure whether the FluidNet baseline is used in the correct way (just using a model pre-trained on a different matrix does not the intended use)

**Suggestions and questions:**

Abstract:
- "require many millions of unknowns" seems fuzzy to me, I would use different wording that "require" here
- in general the abstract could be made more concise
- mention some more relevant quantitative data points in the abstract such as the actual computational cost involved in training the inverse model (not just iteration count)
- explain why a neural network approach is helpful when the inverse is a linear function (or is it a linear neural network)?
- explain the connection between SGD updates and Krylov subspace updates and the information complexity involved (why can the data be more efficient than the Krylov updates)
- obvious necessary improvement would be to consider different sources of linear systems (or otherwise make the title of the paper more specific and pick a different target publication venue)

Introduction:
- "since this decreases the degree of nonlinearity" what is meant by nonlinearity here?
- "We use unsupervised learnin" how is this unsupervised learning? There is "supervision" from A, right?
- Do they account for changing A in every iteration of a PDE solver? Or is the approach mostly suitable for repeatedly solving with the same A?
- How does the approach really generalize to unseen A?
- Learning or computing a perfect inverse for a given A would in principle solve the problem perfectly, but that is not a very useful insight.

Related work:
- "However, in computer graphics applications, at least four orders of magnitude [...]" would be good to have references for these claims.
- The figure is not really related to the method explained in the paper. I would rather use an illustration the demonstrates the workflow or method of the paper.

Motivation:
- For the paper targeting an ML conference I would say providing the PDE and math related to the problem is not really that helpful. I would rather focus on explaining the resulting structure of the matrix. But explaining just the discretization scheme does not tell the reader a lot about how the matrix will look like.

Method:
- Is there going to be an explanation on the effect of their pre-conditioner approach on the eigenspectrum of the matrix (as this reflects conditioning / convergence properties of CG)?
- Why not a linear function for the pre-conditioner network?
- Explain where the network is trained in relation to the PDE solver earlier in the paper

Architecture:
- "Our approach has modest training requirements" what does that mean quantitatively? Also the rest of the sentence, these statements are too vague to be meaningful.
- Again, need to discuss whether a network trained for a given matrix for a given set of boundary / initial conditions is in any way useful to another problem
- Again, not sure why the approach is called unsupervised, the Matrix vector product with a given A is a form of supervision
- Explain earlier and clearly where the "training input vectors" come from for you network training
- Isn't getting approximate eigenvector for A just as expensive as solving the system? When reading that these are used I can imagine that they will be very helpful in approximating and inverse because finding them is essentially as hard as finding the inverse.
- What does it mean to "effectively approximate the full spectrum"?
- There should be much more focus on the computational cost and tradeoff of these aspects of how to compute the eigenvectors etc., which ones to compute (computing them for different parts of the eigenspectrum can have different difficulty, depending on the eigenvalue distriubtion).
- Does m=10k work equally well for all different kinds of n? That seems somewhat implausible? m should probably depend on the discretization? Are the experiments run on m as hyperparam presented in the work? Is the cost of computing m eigenvectors part of the computational cost comparison? What type of pre-conditioner could be constructed with m=10k eigenvectors explicitly?
- What is the motivation for the 9 constant in the definition of c_j^i? Where does theta=500 come from?
- Need to discuss exactly what is the cost of the m Lanczos iterations? That seems like information that would typically be usable to kick start the system solve for A as well.
- The concrete numbers used for the grid sizes and so on points to the fact that this paper is quite specific to a given application and should possibly be titled as such

Model Architecture:
- Would be helpful to get some intuition on why convolutions may be suitable for the type of A. A guess would be that since PDE solutions are smooth and neighboring entries in A related to spacial neighborhood in the discretization grid somehow the convoultion helps with generating smoother pre-conditioners in that same sense? What does the inverse of A look like in terms of "smoothness" structure? Does convolution hurt sparsity?


**Summary Of The Paper:**

The authors propose a method for "learning" a pre-conditioner network for the conjugate gradient method as it applies to linear systems Ax = b resulting from the discrete Poisson equations.
The authors propose to train a CNN model to approximate the inverse of a given matrix A (modulo scaling) by using training data generated from linear combinations of eigenvectors of A which need to be computed first.
Using the proposed method in the context of a conjugate gradient solver shows a significant reduction in the number of iterations required to reach a certain residual norm for their specific test application (a wall-clock time reduction is indicated for larger grid sizes).

**Summary Of The Review:**

I do not recommend to accept the paper in its current form since the specific application does not necessarily seem relevant to the audience of ICLR and otherwise the analysis of the approach is still a little bit too superficial to warrant publication. The paper could be improved by extension to other types of matrices, a more in depth analysis of the presented method, e.g. how and what the pre-conditioner network learns based on e.g. what parts of the eigenspectrum it sees, a better explanation of the total computational costs and also computational complexities involved in the compared approaches and perhaps a more comprehensive treatment of baselines.

---

> ### Author Response · Authors · 2022-11-19
> **Response to Reviewer  gFH2  (1/4)**
>
> We thank the reviewer for the time and effort conducting this thorough review. We briefly address each question below.
>
> **Strengths And Weaknesses:**
>
> **1. The paper addresses a very specific problem type and is not as general as the title suggests.**
>
> We refer the answer for Q10 in Architecture section.
>
> **2.For someone not familiar with the literature on approximating matrix inverses the paper could improve on motivation for the network architecture, for example why are non-linearities even necessary / helful if the learned function is linear.**
>
> We note that finding the inverse of a matrix A is nonlinear, which is why using a CNN for such a problem can potentially be beneficial over standard techniques. For the model model architecture choice, briefly: we first tried a simple deep feedforward convolutional network, which performed adequately but required a large number of layers (i.e., training was inefficient).  Then, motivated by well-known papers like  https://openaccess.thecvf.com/content_cvpr_2016/papers/He_Deep_Residual_Learning_CVPR_2016_paper.pdf, we considered using residual connections with large numbers of layers.  To reduce training and inference time, we found that creating parallel layers of CNNs with downsampling reduced the number of layers required.  In summary, our goal was to first identify the simplest network architecture that provided adequate accuracy for our target problems, and subsequently, we sought to make architectural changes to minimize training and inference time.  We are interested in a more thorough ablation study of potential network architectures, filter sizes, etc., to better characterize the tradeoff curves between accuracy and efficiency; in fact, we believe that substantial further optimization could be done to the network and that our architecture is not too specific to the nature of the problem.
>
> **3.The paper does not contain a theoretical analysis of the computational complexity of obtaining eigenvalue and matrix inverse approximations. To me, it is not clear why a method that uses stochastic gradients (Adam) should converge to a good matrix inverse approximation faster than more specific deterministic approaches, especially if you already have a sample of eigenvectors available.**
>
> Although there might be better matrix inverse approximations given the eigenvectors, that approximation requires much more complexity than a single inference. E.g., given eigenvectors ${v_1,v_2,...,v_k}$ with eigenvalues $\{\lambda_1,\lambda_2,...,\lambda_k\}$ , one can solve $Ax=b$ via $x = v_1*<v_1,b_1>/\lambda_1+ … +v_k*<v_k,b>/\lambda_k$ , but this requires $k$ vector dot products, which is more costly then single inference $x = model(b)$. Hence DCDM becomes much faster compared to using full eigenspectrum.

---

> > ### Author Response · Authors · 2022-11-19
> > **Response to Reviewer gFH2 (2/4)**
> >
> > **4.The experimental setup is not always well motivated and seems somewhat arbitrary, for example, it seems to me that the choice of m=10k example eigenvectors should be somehow related to the eigenspectrum.**
> >
> > This was indeed chosen experimentally. We tried m=5k, 10k, and 20k, and picked the one with the best result.  We will more thoroughly discuss how we decided on example parameters for our experiments in the revised version.
> >
> >
> > **5.The paper does not consider the computational cost of the whole end-to-end setup. Is the method useful if only a single solve with the matrix A is performed given how expensive it is to generate the training set and train the network?If we require multiple solves with the e.g. similar matrices A, then how many solves are typical for a given PDE solver and how useful can this method be?**
> >
> > When the simulation is running, each frame creates a domain with different boundary conditions and this creates a linear system with a different matrix corresponding to the boundary condition of the domain.
> > For example, in the last example in the supplementary video, there is a box in a cube, which rotates each frame. This changes the boundary condition for the smoke plume as it passes around the cube.
> > As explained in Sec 3, this domain creates a pressure system matrix, which corresponds to the domain with boundary conditions.
> > This highlights our strongest point of the paper: the use of a single model trained over a simple/``empty” domain can be used for all domains with different internal boundaries/boundary conditions, encountered in the simulation process. (In the paper we used notation A^{(0,1)^3} to represent the empty domain matrix, and A_omega as the matrix encountered during the simulation.) Hence, we did not consider end-to-end setup for time comparison in Table 1, because the training is only done once, and the model is used for various systems indefinitely.
> >
> > And for the second question, you are right that a typical simulation (such as our rotating cube example) will demand solving different—yet similar—matrices on each time step of the simulation (e.g., due to varying internal boundary conditions from moving solids).  This requirement to solve new linear systems on every time step, is the reason that this type of linear algebra problem is a bottleneck in computational fluid dynamics codes, and in turn is why DCDM is such a useful method since we can accelerate this process.
> >
> >
> > **6.What residual norm for the linear system is required in order for the PDE solver to converge?**
> >
> > We used L2 norm, and run DCDM until reaching 4 order of magnitude decrease in the error norm.
> >
> > **7.The paper does not show how well the network model actually learns / generalizes on a validation set and does not compare different architecture options in that way.**
> >
> > For this question, we are referring to our answer to question 2.
> >
> > **8.I'm not sure whether the FluidNet baseline is used in the correct way (just using a model pre-trained on a different matrix does not the intended use)**
> >
> > We compare to examples found in the original FluidNet paper as well as examples that we do not believe are exactly in their dataset (though we emphasize how FluidNet successfully demonstrated generalizability to a variety of problems in their original paper).
> > For the smoke plume example, (the first example of our supplemental video), the boundary condition situation replicates an example from the FluidNet paper and should be included in their dataset.
> > And we note that residuals in the original FluidNet paper always go down around 1-2 orders of magnitude, which is exactly the behavior we see when we run the pre-trained FluidNet.
> > The point is that we actually don’t care whether FluidNet goes down 1 order of magnitude or 4; it is not an iterative method and so only has “one chance” to get a good solution to the linear system, whereas DCDM (being an iterative method like conjugate gradients) can repeatedly improve upon the solution estimate to (in theory) get arbitrarily low errors (given a sufficiently well-trained network).
> >
> >
> > **Related work:**
> >
> > **"However, in computer graphics applications, at least four orders of magnitude [...]" would be good to have references for these claims.**
> >
> > We will add references for these claims in the appendix section in the final revision.
> >
> > **The figure is not really related to the method explained in the paper. I would rather use an illustration the demonstrates the workflow or method of the paper.**
> >
> > If we have enough space, we will update that picture as you said.

---

> > > ### Author Response · Authors · 2022-11-19
> > > **Response to Reviewer gFH2 (3/4)**
> > >
> > > **Architecture:**
> > >
> > > **1."Our approach has modest training requirements" what does that mean quantitatively? Also the rest of the sentence, these statements are too vague to be meaningful. **
> > >
> > > We meant that our training dataset does not require the generation of flow simulation data, e.g. as was required in Tompson et al. 2017. When generating flow simulation data, one has to choose many ad hoc representative boundary conditions which can be non-trivial to design. Furthermore, advection and the other components of a fluid solver must be evaluated. Instead, we only use approximate eigenvectors of the matrix from the ``empty domain,” i.e. with no interior boundary (see Figs. 1 and 2). This is an important distinction relative to prior works. We will clarify these sentences in the revision.
> > >
> > > **2. Again, need to discuss whether a network trained for a given matrix for a given set of boundary / initial conditions is in any way useful to another problem**
> > >
> > > The network we trained already generalizes to a variety of Poisson problems, which is already quite useful for practitioners in computational fluid dynamics. As future work, we believe this method can extend for similar sparse SPD systems. Please refer to the answer for Q10 in the Architecture section.
> > >
> > > **3.Again, not sure why the approach is called unsupervised, the Matrix vector product with a given A is a form of supervision**
> > >
> > > By unsupervised learning, we meant that we are not providing the actual inverses of the given dataset. More precisely, the dataset only consists of the RHS vectors ${b_1,b_2,...,b_k}$, not including the solutions ${y_i = A^{-1}b_i}$. But we understand the concern, this can be generalized as self-supervised, as reviewer 1 suggests.
> > >
> > > **4. Explain earlier and clearly where the "training input vectors" come from for you network training.**
> > >
> > > Actually we do not understand the question/suggestion here. Can you elaborate on this?
> > >
> > > **5.Isn't getting an approximate eigenvector for A just as expensive as solving the system? When reading that these are used I can imagine that they will be very helpful in approximating and inverse because finding them is essentially as hard as finding the inverse.What does it mean to "effectively approximate the full spectrum"? **
> > >
> > > Yes, creating approximate eigenvectors is even harder than solving the system. But, as explained in earlier question 5 at “Strengths And Weaknesses”, this is only required for dataset generation and once for an extremely simple “empty domain” matrix, and then the trained DCDM model/algorithm can be used for arbitrary, different matrices during fluid simulations.
> > >
> > > **6.There should be much more focus on the computational cost and tradeoff of these aspects of how to compute the eigenvectors etc., which ones to compute (computing them for different parts of the eigenspectrum can have different difficulty, depending on the eigenvalue distriubtion).**
> > >
> > >
> > > In our method, computing approximate eigenmodes is required only for dataset generation, using the simple matrix corresponding to the ``empty domain,” with no interior boundary. At the inference phase, we don’t require  computing eigenmodes. Again, please refer to question 5 at “Strengths And Weaknesses”.
> > >
> > > **7. Does m=10k work equally well for all different kinds of n? That seems somewhat implausible? m should probably depend on the discretization? Are the experiments run on m as hyperparam presented in the work? Is the cost of computing m eigenvectors part of the computational cost comparison? What type of pre-conditioner could be constructed with m=10k eigenvectors explicitly?**
> > >
> > > m=10K is most likely not the optimal choice for all n, but it does the best job among m=5k,10k, and 20k for the N=64 model. m should depend on the grid size n, but this paper is lacking detailed investigation on m; we picked m experimentally. Creating m eigenvectors cost is counted in dataset creation, and not accounted Table 1, because, as explained previously, training is only done once for a single matrix, then never again (even by users who download and use our pre-trained DCDM in their codebases). So the cost of training+dataset generation can be disregarded for timing comparisons.
> > >
> > >
> > > **8.What is the motivation for the 9 constant in the definition of $c_j^i$? Where does theta=500 come from?**
> > >
> > > Originally we assumed this from the Riemann-Lebesgue Lemma, which states that the Fourier spectrum of a continuous function will decay at infinity. However, we also observed in practice that the right hand sides of the pressure system that arose in flow problems (in the empty domain) tended to be at the lower end of the spectrum.  $c_j^i =9$ means 90 % of the lower spectrum is dominant. Theta is chosen experimentally.

---

> > > > ### Author Response · Authors · 2022-11-19
> > > > **Response to Reviewer gFH2 (4/4)**
> > > >
> > > >
> > > > **9.Need to discuss exactly what is the cost of the m Lanczos iterations? That seems like information that would typically be usable to kick start the system solve for A as well.**
> > > >
> > > >
> > > > In paper we report at the end of section 5.1 at page 7  that “Dataset creation takes 5–7 hours for a 64^3 computational grid, and 2–2.5 days for a 128^3 grid.” Lanczos iteration takes the majority/huge of the time in dataset generation (more than 80 percent).
> > > >
> > > >
> > > > **10.The concrete numbers used for the grid sizes and so on points to the fact that this paper is quite specific to a given application and should possibly be titled as such.**
> > > >
> > > > Regarding the numbers for used grid size, our paper used the same numbers as e.g. Tompson et al. This is common for graphics and computer vision papers. As we said in section 7, on page 9, Our network was designed for and trained exclusively using data related to the discrete Poisson matrix, which likely limits the generalizability of our present model. However, we believe our method is readily applicable to other classes of PDEs (or general problems with graph structure) that give rise to large, sparse, symmetric linear systems.
> > > >
> > > >
> > > > **Model Architecture:**
> > > >
> > > > **1. Would be helpful to get some intuition on why convolutions may be suitable for the type of A. A guess would be that since PDE solutions are smooth and neighboring entries in A related to spacial neighborhood in the discretization grid somehow the convolution helps with generating smoother pre-conditioners in that same sense?**
> > > >
> > > > That is a good guess for why convolution works, but we believe it is hard to confirm.
> > > >
> > > > **2. What does the inverse of A look like in terms of "smoothness" structure? Does convolution hurt sparsity?**
> > > >
> > > > We do not understand how convolution relates to sparsity. Can you elaborate on this?  In general, the inverse of a sparse matrix A is a totally dense matrix, so sparsity—while ideal—is not a concern when trying to approximate the inverse of a general sparse A (if this is what the reviewer is asking about).
> > > >
> > > > Finally, we thank you for taking the time to review our paper. We would be happy to have any further discussions.

---

> > > > > ### Comment · Reviewer_gFH2 · 2022-11-29
> > > > > **Thank you for the responses**
> > > > >
> > > > > Thank you for providing detailed answers to the individual questions raised in my review. Since the proposed answers do not necessarily address the main reason for my reject recommendation I will not change the score. I still think that the paper should either experiment with different application examples to demonstrate generality and/or show more of the experimental work done to come up with the proposed architecture to get a better understanding what works and doesn't. Like the authors mention in their responses, they plan to do ablations and further optimization in the future, see:
> > > > > > We are interested in a more thorough ablation study of potential network architectures, filter sizes, etc., to better characterize the tradeoff curves between accuracy and efficiency; in fact, we believe that substantial further optimization could be done to the network and that our architecture is not too specific to the nature of the problem.
> > > > >
> > > > > I also feel like there is still not a clear cost comparison and advantage over other methods pre-conditioned conjugate gradient methods. I'm sorry if my questions in this direction were hard to understand. I think that computing a set of eigenvectors yields a certain amount of information about a matrix. If you train a neural network on that information then the neural network will have learned something. I would like to see clear evidence that what the neural network learns is somehow more generalizable or otherwise useful than what could be obtained from the eigenvectors via deterministic methods.

---

### Official Review · Reviewer_GXf5 · 2022-10-21

**Confidence:** 4
**Correctness:** 4
**Technical Novelty And Significance:** 2
**Empirical Novelty And Significance:** 2
**Recommendation:** 3

**Clarity, Quality, Novelty And Reproducibility:**

The manuscript is clear, novel and the results look reproducible.  However, the novelty in the manuscript is in a direction that does not seem fruitful to me. In particular, the manuscript only demonstrates an improvement over CG for one example problem.

**Strength And Weaknesses:**

The most important thing about iterative methods such as conjugate gradient (CG) for discrete Poisson equations is preconditioning.  By making the step direction selected from a deep learning approach, there is no longer any theory. In particular, there is no underlying polynomial approximation problem that CG is solving, i.e., min_{p in P_k, p(0) = 1} max_{lambda = eig of A}|p(lambda)|. Therefore, one cannot get any idea of how to successfully precondition A, which for this iterative method to be practical on a range of problems will be necessary. For the discrete Poisson equations, the authors show that one can improve the selection of search direction after doing lots of matrix-vector products.

I suspect that the authors have ignored the tensor-product structure in the matrix A^{(0,1)^3}.

The network is designed for and trained exclusively using data related to a single discrete Poisson matrix, which likely limits the generalizability of the present model. The multigrid method is a more general and extremely fast solver for discrete Poisson equations.

**Summary Of The Paper:**

The paper presents an iterative method to solve large sparse pos. def. linear systems of equations with a deep learning technique to select step sizes. The authors demonstrate the efficacy of their approach on linear systems that arise from incompressible flow applications, which are after standard Chorin's splitting end up being shifted discrete Poisson equations. The paper uses deep learning to improve the quality of the search directions that appear in the conjugate gradient method.

**Summary Of The Review:**

The authors treat the search direction in CG as a kind of hyperparameter that can be selected by a deep learning approach. This means that there is no way to mathematical understanding of this iterative method. This makes it impossible to design good preconditioners so the approach is unlikely to generalize or be understanding from a mathematical perspective.

---

> ### Author Response · Authors · 2022-11-19
> **Response to Reviewer GXf5**
>
> We thank the reviewer for their extremely helpful and detailed feedback.
>
> We would like the reviewer to consider our paper as an improvement of the FluidNet paper (Tompson et al., 2017), which is a highly-accepted paper among both the machine learning and computational fluid mechanics communities. We realize that our model and training dataset only studies the Poisson matrix (like Tompson et al.), but underscore that Poisson matrices are often the bottleneck in computational fluid dynamics codes, and hence our algorithm already has practical applicability.  Moreover, as mentioned in the last section of the paper, we believe that our method could apply to other classes of matrices as well.  Although we do not include rigorous bounds on network performance in this paper (similar to other papers like FluidNet), we remark that this is a significant further contribution and one we aim to pursue in future work.  Nonetheless, we also note that many preconditioners—even rigorously-derived algebraic multigrid methods—have numerous parameters that practitioners often tune by hand while experimenting with their problems.  Finally, although we do not have rigorous bounds for our method, DCDM is derived from a careful mathematical understanding of conjugate gradients and should not be viewed as something like a black-box linear solver replacement.
> Finally, we thank you for taking the time to review our paper. We would be happy to have any further discussions.

---

### Official Review · Reviewer_2Pr7 · 2022-10-23

**Confidence:** 4
**Correctness:** 4
**Technical Novelty And Significance:** 2
**Empirical Novelty And Significance:** 2
**Recommendation:** 5

**Clarity, Quality, Novelty And Reproducibility:**

I believe the paper sometimes lacks clarity (see above for points on notation and contribution). The algorithm details are provided however a link to existing codebase would aid in reproducibility.

**Strength And Weaknesses:**

Below are my opinions that would strengthen the paper:

- The will benefit from a clearly outlined contribution section.

- Terms in the paper are used before explaining what they mean (eg., A-conjugate (in page 2), A-orthogonal (in page 4)). The paper would benefit from a notation section. What does $(0,1)^3$ mean in the paper?

- Is the number of iterations required to find a solution using DCDM bounded (similar to CG and Krylov subspace methods)? An analysis on the convergence of DCDG is lacking. What conditions on the trained network would be needed for such an analysis to be feasible?

- The linear system considered (please correct me if I am wrong) in the paper requires the matrix $A$ to have entries in the interval $(0,1)$. Is the algorithm limited to such matrices or can it be generalized larger intervals? If so, why?

**Summary Of The Paper:**

In this paper, the authors consider the general problem of solving a linear system of equation in the case that the matrix is positive semi-definite (and sparse). The authors study a data-driven variation of conjugate gradient to solve the linear system. The algorithm (DCDM) is an iterative algorithm inspired by conjugate gradient algorithm where the descent direction is computed using a trained convolutional neural network. In the paper, the authors specialize their presentation of the algorithm for the incompressible flow problem. Comparison of performance DCDM, CG and preconditioned CG is provided.

**Summary Of The Review:**

I believe the main idea in the paper is interesting however the paper lacks clarity and the contributions of the work is not clear. An analysis of convergence of the algorithm is also needed. For these reasons, I believe the paper is marginally below the threshold for acceptance.

---

> ### Author Response · Authors · 2022-11-19
> **Response to Reviewer 2Pr7  (1/2)**
>
> We thank the reviewer for their extremely helpful and detailed feedback; we provide responses below (in the order of the review sections).
>
> **Strength And Weaknesses:**
> **1. They will benefit from a clearly outlined contribution section.
> Terms in the paper are used before explaining what they mean (eg., A-conjugate (in page 2), A-orthogonal (in page 4)). The paper would benefit from a notation section. What does $A^{(0,1)^3}$ mean in the paper?**
>
> As suggested, we will add a notation section in the appendix of the final version. $A^{(0,1)^3}$ represents the Poisson matrix corresponding to an empty domain.
> When the simulation is running, each frame creates a domain with different boundary conditions, creating a linear system with a different matrix corresponding to the boundary condition of the domain. For example, in the last example in the supplementary video, there is a box in a cube, which rotates each frame. This changes the boundary conditions for the smoke plume as it passes around the cube. As explained in Sec 3, this domain creates a pressure system matrix , which corresponds to the domain with boundary conditions.
> We acknowledge that the notations and definitions are confusing for people who are outside of Computational Fluid Dynamics (CFD) community. We will add a careful appendix and notation section to the paper.
>
> **2. Is the number of iterations required to find a solution using DCDM bounded (similar to CG and Krylov subspace methods)? An analysis on the convergence of DCDG is lacking. What conditions on the trained network would be needed for such an analysis to be feasible?**
>
> We want to explore this very good question, but we do not have the exact answer yet. In our research, we encountered models with different architectures, or trained over different datasets, so that the DCDM does not converge at all or converges up to some point (e.g., converges only two orders of magnitude, then stops converging). Due to page limits, we could add a detailed discussion of the ablation study, but this question is in the direction we want to explore further in our upcoming papers. In addition, there is interesting theoretical follow-up work to derive convergence guarantees for the network/method, and we will emphasize this more in our discussion of future work. However, a proper rigorous analysis will need to come in a future paper.
>
> **3. The linear system considered (please correct me if I am wrong) in the paper requires the matrix A to have entries in the interval (0,1)**
>
> The matrix A consists of integers ranging from integers -6 to 3. Note that this is for 3D. An example of how to create a matrix from the domain is shown in Figure 1. This comes from the standard discretization of the Poisson equation, commonly used in the computational fluid dynamics community.
>
> **4. Is the algorithm limited to such matrices or can it be generalized larger intervals? If so, why?**
>
> We are very interested in this question. In Section 7, we mention that our network was designed for and trained exclusively using data related to the discrete Poisson matrix, which likely limits the generalizability of our present model. However, we believe our method is readily applicable to other classes of PDEs (or general problems with graph structure) that give rise to large, sparse, symmetric linear systems.  We have not thoroughly tested other classes of matrices besides Poisson systems, but it seems like as long as A*x is cheap to compute, there is a good possibility that DCDM will be able to generalize (given appropriately more diverse training data).  We hope to pursue this avenue of inquiry in future work.

---

> > ### Author Response · Authors · 2022-11-19
> > **Response to Reviewer 2Pr7 (2/2)**
> >
> > **Summary Of The Review:**
> >
> > **I believe the main idea in the paper is interesting however the paper lacks clarity and the contributions of the work is not clear. An analysis of convergence of the algorithm is also needed. For these reasons, I believe the paper is marginally below the threshold for acceptance.**
> >
> > As a summary, we want to emphasize our strongest point of the paper: the use of a single model trained over a simple/``empty” domain can be used for all domains with different internal boundaries/boundary conditions. As explained above, each frame in the simulation creates a different Pressure system. State-of-the-art methods require recomputing the preconditioner each time following the matrix changes. In contrast, our model is trained over  a single matrix $A^{(0,1)^3}$ but is used for different matrices (corresponding to domains with different boundary conditions) with the DCDM algorithm.
> > We also refer our answer for question two for convergence analysis.
> >
> > Finally, we thank you for taking the time to review our paper. We would be happy to have any further discussions.

---

### Official Review · Reviewer_GZHi · 2022-10-23

**Confidence:** 4
**Clarity, Quality, Novelty And Reproducibility:** All good.
**Correctness:** 2
**Technical Novelty And Significance:** 3
**Empirical Novelty And Significance:** Not applicable
**Recommendation:** 3

**Strength And Weaknesses:**

Strengths:
1) The authors present an interesting data-driven approach to solve elliptic SPD linear systems. To the best of my knowledge, the method is novel. The training stage is described sufficiently well.
2) The experiments suggest that the scheme works.
3) The paper is clear and well-written.
4) The supplemental material is very good (codes+video).

Weaknesses:
1) The proposed scheme appears to be impractical. Even with the best case settings, training takes from 20 minutes to 120 minutes (for the two smaller grids). These timings are nowhere on Table 1. What about the 256 grid? Also, do these timings include the Lanczos phase or only the training phase after the eigenmodes are computed? I suspect Lanczos is not included but I would like the authors to clarify this.

2) In Table 1, the authors avoid to compare the solution phase of their method against ICPCG, which is nonetheless included later on in Figure 4, but this time they do not report timing results. Please report timings for ICPCG as well. Also, since the problem is elliptic, AMG (algebraic multigrid) is probably a better choice to compare against as this is the state-of-the-art method currently used. Please add this option in addition to Incomplete Cholesky. Based on the timings the authors report, the proposed scheme is slower even compared to plain CG, let alone state-of-the-art AMG preconditioned CG. Indeed, even if I count only two epochs, 60 minutes each, DCDM requires 7200+22 seconds versus 24 seconds for CG. Why is DCDM useful? Because the solution phase is faster? This portion of DCDM reported seems to be tiny compared to the overall running time of the scheme if I count Lanczos+Training. Am I missing something?

3) Even if DCDM was faster with the training time included, it is still extremely impractical as it requires the computation of a huge number of eigenmodes just for the training phase. What is the wall-clock time required to perform 10000 Lanczos iterations? Is this time included in the results shown?
This is a /*huge*/ amount of Lanczos vectors required (and eigenmodes computed). The amount of memory required to save the Lanczos vectors is several orders of magnitude more than that of ICPCG which requires only storage for about six vectors plus the preconditoner (since the authors use Incomplete Cholesky this is pretty low).

4) Please report all timings in the revised version.

Minor comment: DeflatedPCG should be renamed as DeflatedCG, as the former indicates that deflation is applied on an already preconditioned variant of CG, which is not the case here.

**Summary Of The Paper:**

This paper proposed a data-driven modification of the CG algorithm for the solution of SPD linear systems. The proposed algorithm updates the approximate solution using a forward pass of a neural network which is trained with some of the lowest eigenmodes of
a discretization of the elliptic operator without internal boundary conditions. Numerical experiments demonstrate the effectiveness of the proposed scheme in terms of number of iterations and solution runtime.

**Summary Of The Review:**

While I like the topic and the initiative by the authors, and I would like to give a higher score, the proposed method is simply impractical. Unless the authors can show that the timings of Lanczos+Training+Solution is lower than that of ICPCG, there is no reason for a scheme as the one proposed in this paper. Moreover, the memory requirements seem to be enormous compared to ICPCG. Not everyone has access to a 512GB RAM machine. To the latter, please add the need to store CG vectors to enforce A-orthogonality (which can be expensive in terms of both memory and computations).

---

> ### Author Response · Authors · 2022-11-19
> **Response to  Reviewer GZHi  (1/2)**
>
> We thank the reviewer for the time and effort conducting this thorough review.
> We first emphasize our strongest point of the paper: the use of a single model trained over a simple/``empty” domain can be used for all domains with different internal boundaries/boundary conditions.
> To be clear, when the simulation is running, each frame creates a domain with different boundary conditions, and this creates a linear system with a different matrix corresponding to the boundary condition of the domain. State-of-the-art methods require recomputing the preconditioner each time,corresponding to the changing matrix.
> In contrast, our model is trained over a single matrix (corresponding to empty domain Poisson matrix), but is used for different matrices (corresponding to domains with different boundary conditions).
> We note that the concerns with Section 5.1 were due to length constraints and we will improve on this in the final draft in the appendix section. The length constraint also restricted our experimental validation to some degree. We briefly address each question below.
>
> **Strengths And Weaknesses:**
>
> **1. The proposed scheme appears to be impractical. Even with the best case settings, training takes from 20 minutes to 120 minutes (for the two smaller grids). These timings are nowhere on Table 1. What about the 256 grid? Also, do these timings include the Lanczos phase or only the training phase after the eigenmodes are computed? I suspect Lanczos is not included but I would like the authors to clarify this.**
>
> The timings 20 mins and 120 mins only includes training and does not include the dataset generation (which includes Lanczos iteration). Lanczos iteration is required only for dataset generation to calculate the approximate eigenmodes of a single empty domain with no interior boundary. Since training is done only in a single empty domain with no interior boundary, Lanczos iteration is only done once as well for dataset generation. To clarify: someone could download our pretrained models and use them to solve new, previously unseen Poisson systems in a production codebase; neither training nor dataset generation ever needs to be repeated.
> Moreover, we find that the weights and biases of the trained model for lower resolution grids can also be used for higher resolutions.
> For the 256 grid, no new training was done; instead, we used the weights from 128 (and 64) grids. This can be seen in Table 1, e.g., in column 3 (256 grid), row one (DCDM-64) represents the DCDM test for 256 grid, where the model weights are obtained from 64 grid training.
>
> **2. In Table 1, the authors avoid to compare the solution phase of their method against ICPCG, which is nonetheless included later on in Figure 4, but this time they do not report timing results. Please report the timings for ICPCG as well. Also, since the problem is elliptic, AMG (algebraic multigrid) is probably a better choice to compare against as this is the state-of-the-art method currently used. Please add this option in addition to Incomplete Cholesky. Based on the timings the authors report, the proposed scheme is slower even compared to plain CG, let alone state-of-the-art AMG preconditioned CG.Indeed, even if I count only two epochs, 60 minutes each, DCDM requires 7200+22 seconds versus 24 seconds for CG. Why is DCDM useful? Because the solution phase is faster? This portion of DCDM reported seems to be tiny compared to the overall running time of the scheme if I count Lanczos+Training. Am I missing something?**
>
> The reason for skipping the time comparison for ICPCG in Table 1, is that as we said at last of chapter 6 on page 9,  Although ICPCG successfully reduces the number of iterations Figure 4, 4b, we found the runtime to scale prohibitively with grid resolution, so we exclude it from comparison in Table 1.
> For example, incomplete Cholesky factorization as a preconditioner for 64 grid (which corresponds to 262144 by 262144 matrix) takes more than 2 days. As we explained previously, we only do the training for the matrix associated with the empty domain. We do not perform Lanczos iteration nor train on any domain with internal boundaries. We again refer the reviewer end of Section 3 and Section 5 for discussion of these points. Hence for timing, we did not include training+dataset creation.These operations only ever need to be performed once; an end user can download our pre-trained models and immediately solve new, previously unseen Poisson systems, with absolutely no new training or dataset generation (just model inference, cf. Table 1).
> So DCDM is useful, because even when the domain changes (i.e. the matrix changes) at each frame, we can use the same model still, instead of creating a new preconditioner each time, which is required in ICPCG or Deflated methods.
> We are happy to add AMG timing comparisons as well for the final version of the paper in the appendix section.

---

> > ### Author Response · Authors · 2022-11-19
> > **Response to Reviewer GZHi (2/2)**
> >
> >
> > **3. Even if DCDM was faster with the training time included, it is still extremely impractical as it requires the computation of a huge number of eigenmodes just for the training phase. What is the wall-clock time required to perform 10000 Lanczos iterations? Is this time included in the results shown? This is a /huge/ amount of Lanczos vectors required (and eigenmodes computed). The amount of memory required to save the Lanczos vectors is several orders of magnitude more than that of ICPCG which requires only storage for about six vectors plus the preconditoner (since the authors use Incomplete Cholesky this is pretty low).**
> >
> > In paper, we report at the end of ch 5.1 on page 7  that “Dataset creation takes 5–7 hours for a 64^3 computational grid, and 2–2.5 days for a 128^3 grid.” Lanczos iteration takes the majority/huge of the time in dataset generation (more than 80 percent). After the training, there is no need to store Lanczos vectors for model to run.
> > Our model also requires less space compared to ICPCG. More precisely, the $L$ and $D$ matrices for $128^3$ take about 18.7MB in scipy.sparse format. Our network can be stored at less than 500KB.  (Also, we note that while we had access to a machine with a lot of RAM, this RAM was not used at all for training the DCDM network, as the network was trained using a standard workstation GPU).  We will note these advantages in the appendix section.
> >
> > **Summary Of The Review:**
> >
> > **While I like the topic and the initiative by the authors, and I would like to give a higher score, the proposed method is simply impractical. Unless the authors can show that the timings of Lanczos+Training+Solution is lower than that of ICPCG, there is no reason for a scheme as the one proposed in this paper. Moreover, the memory requirements seem to be enormous compared to ICPCG. Not everyone has access to a 512GB RAM machine. To the latter, please add the need to store CG vectors to enforce A-orthogonality (which can be expensive in terms of both memory and computations).**
> >
> > We believe that the reviewer’s comments about comparisons with incomplete Cholesky preconditioning (ICPCG) are also made without realizing that we only train the network for the empty domain matrix; with one single Lanczos+Training session, we can use DCDM on any frame of a fluid simulation with any type of Poisson matrix that arises (and we have done this; users do not need to retrain anything to start using DCDM). On the other hand, an approximate factorization must be recomputed on every single frame when doing ICPCG with the matrices that arise in a domain with moving internal boundaries. (Note, this is also the case using the approach of Tompson et al. 2017.)  We hope this clarifies the advantage of our method and why our timings are compared fairly against methods like ICPCG.
> > And as shown in fig 4d, DCDM does not require full A-orthogonality. Hence the algorithm only stores two previous vectors, the same as the standard CG algorithm. We will highlight this more clearly in the revised manuscript.
> >
> > Finally, we thank you for taking the time to review our paper. We would be happy to have any further discussions.

---

> > > ### Comment · Reviewer_GZHi · 2022-11-28
> > > **Discussion**
> > >
> > > Thank you for providing these additional details. While my initial comments maybe did not make this apperent, I was aware of the above remarks from my initial reading of the paper. I am in total agreement with the authors regarding why an algorithm such as the one proposed in this submission is useful; the main issue though remains. The pre-processing time (time to generate the dataset + perform training) is extremely long, and requiring the storage of thousands of Lanczos vectors is not practical. Note that I am not debating the method itself (other than the Lanczos part), but rather its illustration.
> > >
> > > In fact, to keep my response short, I was looking for a very simple comparison. What is the wall-clock time if:
> > > a) I use the proposed approach (including all of: dataset generation, training, and solving each linear system),
> > > b) I use AMG for each frame separately, by using the solution of the previous frame as my initial approximate
> > > solution, and
> > > c) same as "b)", but now with ICPCG with progressively lower drop-tolerance.
> > >
> > > If "a)" is always faster, then I think that the proposed approach is certainly useful. Note that one could do even better
> > > than "b)" and "c)" if recycles the eigenvectors associated with the smallest eigenvalues of the initial frame matrix.
> > > I also suggest the authors the following paper:
> > > https://epubs.siam.org/doi/abs/10.1137/S1064827598310227
> > >
> > > Memory-wise, I am a bit skeptical about the argument that the model requires less storage than ICPCG (a light-weight preconditioner).
> > > The algorithm requires the use of Lanczos iteration to generate the dataset; therefore, this cost must be added to the storage requirements.
> > > To see why I do not agree with the storage-complexity note, consider the following statement: "The cost to solve a general linear system through LU is quadratic (with respect to 'n') because this is the complexity of a triangular substitution; even though the cost to obtain the LU decomposition is cubic". Do you agree? I suggest the authors to consider using a lower threshold in ICPCG and see how their method compares against it.

---

### Official Review · Reviewer_nmQe · 2022-10-27

**Confidence:** 4
**Correctness:** 3
**Technical Novelty And Significance:** 4
**Empirical Novelty And Significance:** 4
**Recommendation:** 8

**Clarity, Quality, Novelty And Reproducibility:**

The paper is very clearly written and as far as I can tell original. Reproducible code is provided in the supplementary material.

**Strength And Weaknesses:**

The idea proposed by the authors is simple, effective, and by my standards the right way to use neural networks to accelerate forward solves. Instead of attempting to directly regress the solution or trying complicated things, which more or less never give reconstructions that are good for applications (and tends to horribly overfit training data), they opt to accelerate an iterative scheme in a new way, by regressing a new descent direction. Although there exist some related ideas, the whole package with the self-supervised training scheme and a well thought-out training set construction is quite novel as far as I can tell.

I further find the idea to be very well executed. The authors clearly have a strong understanding of numerical analysis and numerical linear algebra. The literature overview and the problem description are very clear. The experimental validation is convincing and the reproducible code is provided in the supplementary material. I do have a couple of questions to clarify below but overall my impression is very positive.


- You write that "We only ask that our network approximate the inverse up to an unknown scaling since this decreases the degree of nonlinearity and since it does not affect the quality of the search direction (which is scale independent)." My intuition is that the degree of nonlinearity in the network which computes the descent direction is not important. Rather, the norm of the vector computed by the network would almost certainly not be the optimal step length and so it makes sense to compute it exactly as you do in your loss. (That is, even if the network was trained to regress something with a specific norm it would be opportune to rescale it; one might try training like that and rescaling only at test time.)

- I don't think that your training strategy is unsupervised. Rather, the supervision is performed through the operator $A^\Omega$. A commonly used term in this situation where you have the residual both as input to f and as the target is "self-supervised learning".

- The abstract suggests that the number of iterations required by your method is independent of the problem size, but Table 1 indicates otherwise. Could you clarify this?

- I find it a bit surprising that the model from the third epoch was optimal at resolution 128^3. Is "optimal" meant in the training loss sense or in the sense that it performed best at test time? If the latter is the case, does the loss keep decreasing after the third epoch? And if not, can you speculate why?

- What exactly is meant by the "data dependency" of incomplete Cholesky preconditioners on page 2? Is this alleaviated by the presented method (I am wondering about the word "However, ")


### Minor

- please add callouts (a, b, c, d) in Figure 4 even though they seem obvious
- please use the same color for the same algorithm in all subfigures, otherwise it's quite confusing (you might also consider other ways to vary the line style as well as using a color scheme that is friendly to the color blind---red / green is quite bad)




**Summary Of The Paper:**

The authors propose a machine-learning method to accelerate conjugate descent by training a convolutional network to regress good descent directions. The central idea---to obtain conjugate directions via self-supervised learning---is simple, creative, and it works very well. The authors effectively combine a number of techniques from numerical analysis and numerical linear algebra. They apply their method to build a fast Poisson solver and show that it consistently outperforms principled solvers as well as recent ML-accelerated solvers on realistic benchmarks.


**Summary Of The Review:**

As stated above, I think this is a good paper. It addresses an important problem, finds an effective way to use machine learning (via self-supervision), and does exactly what it says in the introduction, without overselling or underselling. I would be happy to see this paper accepted.

---

> ### Author Response · Authors · 2022-11-19
> **Response to Reviewer nmQe**
>
> We thank the reviewer for their extremely helpful and detailed feedback; we provide responses below (in the order of the review sections).
>
> **Strength And Weaknesses:**
>
> **1. You write that "We only ask that our network approximate the inverse up to an unknown scaling since this decreases the degree of nonlinearity and since it does not affect the quality of the search direction (which is scale independent)."
> My intuition is that the degree of nonlinearity in the network which computes the descent direction is not important.
> Rather, the norm of the vector computed by the network would almost certainly not be the optimal step length and so it makes sense to compute it exactly as you do in your loss. (That is, even if the network was trained to regress something with a specific norm it would be opportune to rescale it; one might try training like that and rescaling only at test time.)**
>
> We agree with your intuition and will note this in the camera-ready version.
>
> **2. I don't think that your training strategy is unsupervised. Rather, the supervision is performed through the operator $AΩ$. A commonly used term in this situation where you have the residual both as input to f and as the target is "self-supervised learning".**
>
> By unsupervised learning, we meant that we are not providing the actual inverses of the given dataset. More precisely, the dataset only consists of the RHS vectors ${b_1,b_2,...,b_k}$, not including the solutions ${y_i = A^{-1}b_i}$. But, you are right, we can say self-supervised.
>
> **3. The abstract suggests that the number of iterations required by your method is independent of the problem size, but Table 1 indicates otherwise. Could you clarify this?**
>
> As you mentioned, the number of iterations is not exactly the same for each problem size. We intended to convey a small number of iterations in terms of problem size compared to traditional methods. We will update these sentences in detail in our revision, thanks.
>
> **4. I find it a bit surprising that the model from the third epoch was optimal at resolution 128^3. Is "optimal" meant in the training loss sense or in the sense that it performed best at test time? If the latter is the case, does the loss keep decreasing after the third epoch? And if not, can you speculate why?**
>
> As you mentioned, “optimal” meant here that it performed best in terms of minimizing the number of iterations required for DCDM to converge when applied to linear systems, i.e., we optimized for how well DCDM actually works as opposed to minimizing a training or test loss. We did observe that loss keeps decreasing after the third epoch, but we did not observe better performance in terms of the actual number of iterations required for the iterative linear solver to converge.
>
> **5. What exactly is meant by the "data dependency" of incomplete Cholesky preconditioners on page 2? Is this alleaviated by the presented method (I am wondering about the word "However, ")**
>
> “however, their inherent data dependency” means that the ICPCG operation is using sparse data structures as a result of incomplete Cholesky decomposition. But accessing this structure (for backward/forward substitution) is difficult to make parallel to speed up operations. Mathematically, each iteration of Cholesky factorization depends on the previous iteration, so parallel speedups are challenging to achieve (though there is some active research in this area).
>
> **Minor:**
> * **please add callouts (a, b, c, d) in Figure 4 even though they seem obvious**
> * **please use the same color for the same algorithm in all subfigures, otherwise it's quite confusing (you might also consider other ways to vary the line style as well as using a color scheme that is friendly to the color blind---red / green is quite bad)**
>
> Thanks for recommending these two improvements.  We will incorporate these into the final version.
>
> Finally, we thank you for taking the time to review our paper. We would be happy to have any further discussions.

---

### Decision · Program_Chairs · 2023-01-20

**Decision:**

Reject

**Justification For Why Not Higher Score:**

Mainly the demonstration in the work is not clear cut from storage to timings. An additional numerical evidence could also help convince the broader applicability of the work.

**Justification For Why Not Lower Score:**

N/A

**Metareview: Summary, Strengths And Weaknesses:**

The authors propose to choose the search direction in conjugate gradients (CG) method by means of a neural network via self-supervised learning. It builds upon Tompson et al 2017 and provides one numerical evidence to support the performance of the new idea.

While the idea is quite nice, its broader applications are questionable as there is not much theory. The architecture is designed specifically using data related to a single discrete Poisson matrix, limiting the generalizability. One of the key ideas in CG, namely preconditioning, is no longer directly useable. There are some some other impractical aspects, such as storing a large number of Lanczos vectors (as opposed to storing a limited number of directions and a preconditioner). As one of the reviewers also mention that the comparisons avoid timings of the preprocessing as well as the storage issues.

To make the paper better, the AC suggests authors to clearly address the weaknesses above, from explicit storage comparisons to total time comparisons.